# Family-based exome sequencing combined with linkage analyses identifies rare susceptibility variants of *MUC4* for gastric cancer

**Yoon Jin Choi**[1☻], **Jung Hun Ohn**[1☻], **Nayoung Kim**[1,2,3]*, **Wonji Kim**[4], **Kyungtaek Park**[5], **Sungho Won**[5,6,7], **Lee Sael**[8], **Cheol Min Shin**[1], **Sun Min Lee**[1], **Sejoon Lee**[9], **Hyun Joo An**[10], **Dong Man Jang**[3], **Byung Woo Han**[3], **Hye Seung Lee**[9], **Seung Joo Kang**[11], **Joo Sung Kim**[2,11], **Dong Ho Lee**[1,2,3]

**1** Department of Internal Medicine, Seoul National University Bundang Hospital, Seongnam, South Korea, **2** Department of Internal Medicine and Liver Research Institute, Seoul National University College of Medicine, Seoul, South Korea, **3** Tumor Microenvironment Global Core Research Center, College of Pharmacy, Seoul National University, Seoul, South Korea, **4** Channing Division of Network Medicine, Department of Medicine, Brigham and Women's Hospital and Harvard Medical School, Boston, MA, United States of America, **5** Interdisciplinary Program of Bioinformatics, Seoul National University, Seoul, South Korea, **6** Department of Public Health Sciences, Seoul National University, Seoul, South Korea, **7** Institute of Health and Environment, Seoul National University, Seoul, South Korea, **8** Department of Artificial Intelligence and Data Science, Ajou University, Seoul, South Korea, **9** Department of Pathology, Seoul National University Bundang Hospital, Seongnam, South Korea, **10** Graduate School of Analytical Science and Technology, Chungnam National University, Daejeon, South Korea, **11** Department of Internal Medicine and Healthcare Research Institute, Healthcare System Gangnam Center, Seoul National University Hospital, Seoul, South Korea

☻ These authors contributed equally to this work.
* nakim49@snu.ac.kr

**Data Availability Statement:** Datasets of 55 enrollees were deposited in the European Nucleotide Archive (http://www.ebi.ac.uk/ena/data/

## Abstract

Genome-wide association studies of gastric cancer (GC) cases have revealed common gastric cancer susceptibility loci with low effect size. We investigated rare variants with high effect size via whole-exome sequencing (WES) of subjects with familial clustering of gastric cancer. WES of DNAs from the blood of 19 gastric cancer patients and 36 unaffected family members from 14 families with two or more gastric cancer patients were tested. Linkage analysis combined with association tests were performed using Pedigree Variant Annotation, Analysis, and Search Tool (pVAAST) software. Based on the logarithm of odds (LOD) and permutation-based composite likelihood ratio test (CLRT) from pVAAST, *MUC4* was identified as a predisposing gene (LOD *P*-value = 1.9×10$^{-5}$; permutation-based *P*-value of CLRT $\leq$ 9.9×10$^{-9}$). In a larger cohort consisting of 597 GC patients and 9,759 healthy controls genotyped with SNP array, we discovered common variants in *MUC4* regions (rs148735556, rs11717039, and rs547775645) significantly associated with GC supporting the association of *MUC4* with gastric cancer. And the *MUC4* variants were found in higher frequency in The Cancer Genome Atlas Study (TCGA) germline samples of patients with multiple cancer types. Immunohistochemistry indicated that *MUC4* was downregulated in the noncancerous gastric mucosa of subjects with *MUC4* germline missense variants, suggesting that loss of the protective function of *MUC4* predisposes an individual to gastric

view/PRJEB29071, accession number:
PRJEB29071)

**Funding:** This work was funded by a grant from the
National Research Foundation (NRF) of Korea to
the Global Core Research Center (GCRC) funded by
the Korean government (MSIP) (No. 2011-
0030001). The funders had no role in study design,
data collection and analysis, decision to publish, or
preparation of the manuscript.

**Competing interests:** The authors have declared
that no competing interests exist.

cancer. Rare variants in *MUC4* can be novel gastric cancer susceptibility loci in Koreans
possessing the familial clustering of gastric cancer.

## Introduction

Gastric cancer (GC) is one of the most common cancers and is the third leading cause of can-
cer mortality worldwide, with an estimated 783,000 deaths in 2018 [1]. South Korea has the
highest incidence of stomach cancer in the world [1]. A positive family history is a well-known
risk factor for GC, in addition to male sex, *Helicobacter pylori* infection, smoking, and frequent
consumption of salty food and dietary nitrite [2]. Most GC cases are sporadic, with approxi-
mately 90% developing in communities carrying only an average risk [3, 4]. Hereditary cancer
syndromes including hereditary diffuse gastric cancer (HDGC) account for less than 3% of all
GC cases [5]. The remaining 7% is found in individuals with a positive family history but with-
out a diagnosed inherited cancer syndrome [5].

Individuals with affected first-degree relatives (FDRs) have a 2 to 3-fold increased risk for
GC [6]. The increased risk in affected families is partly attributable to the sharing of similar
environmental factors, such as dietary habits or *H. pylori* infection. Nevertheless, frequently
observed weak association between *H. pylori* infection and GC development in affected families
[7, 8] suggests a genetic basis for familial aggregation. Given this background, we hypothesized
that genetic predisposition may underlie the high occurrence of GC in GC-prone families.

According to literature, several SNPs associated with GC have been identified via candidate
gene approach [5, 9] or a genome-wide association study (GWAS) [10, 11]. One of the most
well-known is the association of MUC1 with gastric cancer [12]. MUC1 belongs to the mucin
family and it is located at the apical surface of the mucosal epithelial cells and acts as a protective
barrier against exogenous insults. It is hypothesized that MUC1 variants like rs4072037 influ-
ence the quantity and the quality of the MUC1 protein and cause difference in barrier function
in the stomach with subsequent difference in GC susceptibility between individuals [12].

However, studies on such SNPs have yielded inconsistent results, especially in relation to
different GC types and ethnicities [10, 13–15]. Moreover, the effect sizes of SNPs resulting
from GWAS were generally small, less than 2.0. Recently, novel GC genes that explain small
fractions of familial GC have been identified using whole-genome and whole-exome sequenc-
ing (WES), including *PALB2, BRCA1, and CTNNA1* [8, 16, 17]. Most previous studies on
familial clustering of GC have focused on HDGC or diffuse-type GC [8, 16], although a consid-
erable proportion of intestinal-type GC occurs in GC family clusters [7, 18]. WES studies on
GC are rare in Asia, where the prevalence of intestinal-type GC is high.

The objective of this study was to identify GC-associated germline variants with a high
effect size [19] by linkage and association analyses based on WES of subjects with familial clus-
tering of GC not limited to HDGC. We recruited both affected and unaffected family members
and identified *MUC4* as a candidate predisposition gene with a large effect size. We further val-
idated in large populations of cases and controls and through expression analysis of *MUC4* in
normal gastric mucosa and gastric cancer tissues.

## Materials and methods

### Patient inclusion for exome sequencing

From April 2017 to March 2018, GC patients and their FDRs, among families with two or
more members diagnosed with GC within three generations, were enrolled in the study at

Seoul National University Bundang Hospital. Non-GC controls were defined as individuals aged > 50 years with a normal endoscopy within the previous 6 months. For diagnosis of GC, it was based on pathologic diagnosis by endoscopic biopsy or surgical specimens.

Family history of GC, smoking, consumption of alcohol, dietary preference, socioeconomic status, gastrointestinal symptoms and a history of previous *H. pylori* eradication were acquired via questionnaires. Histologic evaluations with Giemsa staining and an anti-*H. pylori* test were performed to determine *H. pylori* infection status [7, 14] (online S1 Table).

All procedures involving human participants were performed in accordance with the ethical standards of the institutional and national research committees and the 1964 Helsinki Declaration. This study was approved by the Institutional Review Board of Seoul National University Bundang Hospital (B-1610-366-303). Members of all families who participated in the present study signed a specific informed consent form.

### DNA isolation and whole-exome sequencing

Genomic DNA was isolated using the Qiagen DNeasy blood and tissue kit(Qiagen, Hilden, Germany) according to the manufacturer's instructions. To perform WES, Agilent SureSelect All Exon V6(Agilent Technologies, Santa Clara, CA) with reagents was used with sequencing libraries and capture. Sequencing was performed on an Illumina HiSeq 2500 platform (2x100 bp-paired end; Illumina, Inc., San Diego, CA). Sequence datasets of the 55 enrollees were deposited in the European Nucleotide Archive (http://www.ebi.ac.uk/ena/data/view) under accession number PRJEB29071.

### Variant detection and annotation

Raw sequencing reads were aligned to the Human Genome Reference Assembly GRCh37/hg19 using Burrows-Wheeler Aligner(BWA v0.7.15) software [20]. The BWA alignment files were converted to BAM files using SAM tools v1.3, and duplicates were marked with Picard (https://sourceforge.net/projects/picard, v1.96). Local realignment, base quality recalibration, and haplotype calling in genomic Variant Call Format (gVCF) mode for each sample were performed using the Genome Analysis Toolkit (GATK v3.5) according to best practices [21]. Genomic VCF(gVCF) files were combined and joint genotyped with GATK. Functional annotation of genetic variants was conducted using ANNOVAR with population frequencies. Variants on Exon 24 of *MUC4* gene were confirmed by Sanger sequencing and all variant reads were inspected with Integrative Genomes Viewer [22].

### Linkage and association analyses

Disease susceptibility loci were identified using linkage analysis and the gene-based association test with Pedigree Variant Annotation, Analysis, and Search Tool (pVAAST) under the autosomal dominant inheritance model and the maximally allowable prevalence of disease, 0.005 [23]. The p values of the LOD (logarithm of odds) scores and a gene-based burden-type composite likelihood ratio test (CLRT) for binomial likelihood based on allele counts in cases and controls weighted by functional prediction were calculated from $10^6$ permuted samples using a gene-drop method. At linkage analysis, we compared variants in 19 affected cases to their unaffected 36 controls in all 14 families considering respective family structures. At the gene-based association test we compared allele counts in whole exomes of 19 GC patients to those of 397 Korean control whole genomes obtained from the National Biobank of Korea, Korea National Institute of Health. A total of 19,491 genes were analyzed, and the Bonferroni-adjusted 0.05 level was $2.57 \times 10^{-6}$. For *MUC4*, $10^8$ permuted samples were used as p value of CLRT score was below $1.0 \times 10^{-6}$ in $10^6$ permuted samples.

## Allele frequency determination

The gnomAD database (http://gnomad.broadinstitute.org/) was used to obtain the frequency of specific variants in overall and East Asian control populations.

## The Cancer Genome Atlas (TCGA) data analysis

TCGA data were downloaded from the Institute for Systems Biology Cancer Genomics Cloud from the Genome Data Commons legacy (GRCh37/hg19) archive and Genome Data Commons Data Portal (https://portal.gdc.cancer.gov). Sequence information was obtained from the database of Genotypes and Phenotypes (dbGaP).

## Genome-wide association of variants located in *MUC4*

Blood samples were collected from 597 histologically confirmed GC patients and 9,758 non-GC subjects who were examined at the Seoul National University Bundang Hospital and the Seoul National University Hospital Healthcare System Gangnam Center. They were genotyped by the Affymetrix Axiom Korean Chip, which consists of 827,783 variants. Any subjects were removed if 1) sex estimated by their genome was different from their clinical information, 2) call rates of subjects were less than 97%, 3) heterozygosity rates deviated by three times their standard deviation from their mean and 4) their identity-by-descent estimates with other subjects were larger than 0.185 and they had higher missing rates than their paired subjects did. Variants were removed whose 1) missing rates were larger than 3% or were significantly different between the case and control groups ($p < 1{\times}10^{-5}$), 2) minor allele frequencies were smaller than 5% and 3) *P* values for the Hardy-Weinberg equilibrium exact test were smaller than 0.001 as suggested by Anderson *et al.* [24]. Then, untyped variants were imputed using the Michigan Imputation Server [25]. After imputation, 4,224 variants located in *MUC4* and its 0.5 MB flanking region were analyzed using logistic regression with adjusting for the effects of sex, age, and top 10 principal components of the sample relationship matrix. The Bonferroni-adjusted 0.05 significance level became $1.18{\times}10^{-5}$. PLINK(v1.90b4.5) and R (v3.5.2) were used for the process [26].

## Immunohistochemistry (IHC) analyses of noncancerous gastric mucosa and gastric cancer tissue

Antral noncancerous mucosa was evaluated using IHC from 15 GC patients and 8 non-GC participants who consented to endoscopic biopsy. In the case of GC patients, cancer tissue was also stained. The antibody for detecting MUC4 (clone: 8G7) (1:100 dilution, Zeta Corporation, Arcadia, CA, USA) was used for IHC. The antibody we have used for IHC detects MUC4α region. The specificity of the antibody was evidenced by previous studies. Overall staining of sections (4 μm thick) was conducted via the BenchMark XT Staining system and ultraVIEW Universal DAB Detection Kit (Ventana Medical Systems, Inc., Tucson, AZ, USA). MUC4 expression was evaluated using light microscopy via multiplication of the intensity by area (%), where staining was observed in the epithelial glands (0 to 300):0, no staining;1+, faint/ barely perceptible partial staining; 2+, weak to moderate staining; 3+, strong staining. In cancerous tissue, only strongly stained area loci were included for scoring. Each sample was scored in a blinded manner by a single pathologist (HSL).

## MUC4 structural computational analysis

Motif search and prediction of peptide cleavage, glycosylation, and protein structure were performed for the analysis of MUC4 structure using the protein sequence (refSeqID:NP_001191215) and mRNA sequence (refSeqID:NM_018406.6), obtained from the NCBI Reference Sequence

Database [27], as references. A motif search was performed using the MotifFinder tool in Geno-meNet(https://www.genome.jp; S1 Fig). Peptide cleavage prediction was performed using Pepti-deCutter [28] on the MUC4 α region of the protein reference sequences (NP_001191215) and variant sequences. NetOGlyc [29] and NetNGlyc (http://www.cbs.dtu.dk/services/NetNGlyc/) were used to predict likely locations of O-GalNAc (N-acetylgalactosamine) and N-GalNAc modifications, respectively. Protein structure predictions using homology modeling of MODELLER [30, 31] and SWISS-MODEL were performed but failed to yield reliable structures at the SNV locus.

## Effect size analysis

The odds ratio (OR) for all significant genes at the Bonferroni-adjusted 0.05 significance level was estimated with logistic regression. Familial correlations were estimated with GMMAT [32], and the variance for the random effect that explains familial correlation was estimated to be 0. Thus, GC status among family members was assumed to be independent, and standard logistic regression was applied using *Rex Version 2.1* (http://rexsoft.org). For each gene, genetic risk scores were coded as 1 if one or more rare alleles in the corresponding gene were observed and 0 otherwise. Sex, age, smoking status and HDGC were included as covariates to adjust for their effects.

# Results

## Characteristics of participants

The total subjects included 55 participants (19 GC patients and 36 non-GC relatives) from 14 independent families (online S1 Table). Pedigrees of the 14 families are presented in Fig 1. Three HDGC families (Nos. 7, 8 and 13) that met the International Gastric Cancer Linkage Consortium 2010 clinical criteria were included [33]. Clinical characteristics of the participants are presented (Table 1; online S1 Table).

The mean age at GC diagnosis was 59.0 years (range: 31–84 years), while that of unaffected relatives was 62 years. A higher percentage of males tended to be present in the GC group than in the other group (63.2% vs. 33.3%, $p = 0.034$). A higher proportion of GC patients experienced smoking than participants in the non-GC group (63.2% vs. 27.8%, $p = 0.011$). Approximately half of GC patients were *H. pylori*-positive, while 75.0% of unaffected relatives were *H. pylori*-positive without significant differences in these proportions. Among the enrolled GC patients, 3 cases were recognized as diffuse-type, 13 cases as intestinal-type, and one case as mixed-type based on Lauren's classification. The specific histologic type of the remaining cases could not be identified even though we requested this information from the hospitals where the patients were treated mainly due to gastric surgery or histology having been performed a long time ago.

## Discovery of germline exome variants associated with gastric cancer

WES data were generated from the 55 subjects with a mean depth of 96-fold on targeted exome regions (online S2 Table). An average of 97% of all targeted regions were covered by at least 20-fold. To explore rare variant candidates for GC, linkage analyses combined with an association test were conducted. Based on LOD *p* values, *MUC4*, *MAGEC1*, and *RETSAT* were identified as putative genes associated with GC (Fig 2A to 2C). At gene-based CLRT analyses that integrate linkage information, case-control association and functional variant prediction, and *MUC4* reached the genome-wide significance level (*p* value $\leq 9.9 \times 10^{-9}$, and the genome-wide Bonferroni-adjusted 0.05 significance level = $2.6 \times 10^{-6}$) (Fig 2A). Fig 2B and 2C show the Manhattan and quantile-quantile (QQ) plots for our linkage analyses, respectively, and they show that our statistical analyses preserve the nominal significance level. S2 Fig shows the QQ

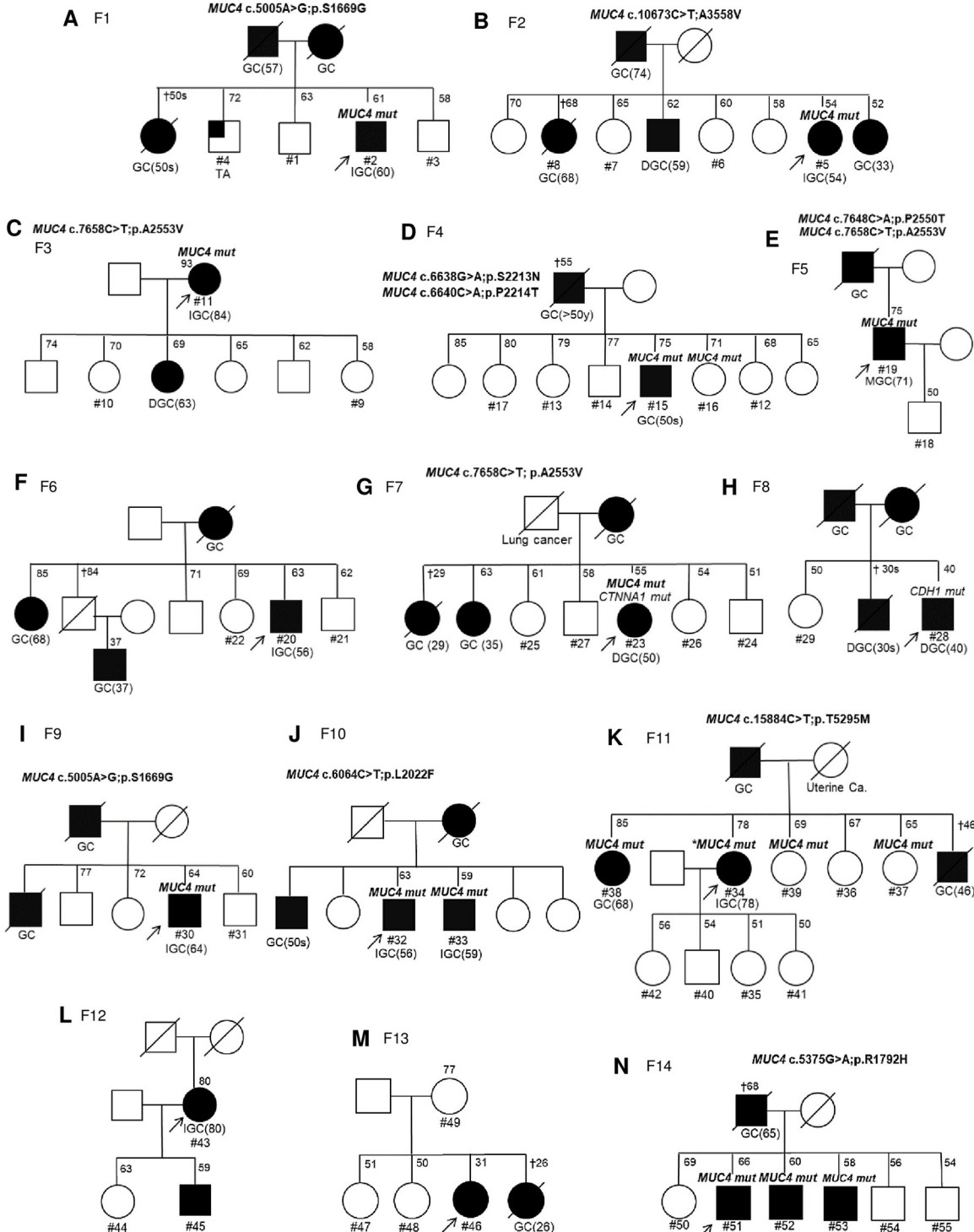

**Fig 1. Pedigrees of families with *MUC4* variants.** Participants whose DNA was analyzed in the present study were given a number beginning with "#." The age at diagnosis of GC is described in parentheses after GC. The arrow indicates a proband. The cross represents death. *mut* denotes a carrier of the variants of genes. \*#34 has another *MUC4* variant, c.5005A>G; p.S1669G. GC, gastric cancer (unknown Lauren classification); IGC, intestinal-type gastric cancer; DGC, diffuse-type gastric cancer; RCC, renal cell cancer; ca, cancer; TA, tubule adenoma; F, family.

**Table 1. Clinical and demographic characteristics of GC patients and non-GC subjects.**

| Variables | Non-GC (*n* = 36) | GC (*n* = 19) | *P*-value[a] |
|---|---|---|---|
| Male | 12 (33.3) | 12 (63.2) | **0.034** |
| Age (years) | 62.14 (9.1) | 59.11 (13.8) | 0.331 |
| *MUC1* (rs4072037) | | | |
| GG | 0 | 2 (10.5) | 0.098 |
| AG | 5 (13.9) | 1 (5.3) | |
| AA | 31 (86.1) | 16 (84.2) | |
| Rural residence | 21 (58.3) | 14 (73.7) | 0.260 |
| Smoking | 10 (27.8) | 12 (63.2) | **0.011** |
| Alcohol consumption | 22 (61.1) | 14 (73.7) | 0.351 |
| Fruit intake ≥3/week | 26 (72.2) | 16 (84.2) | 0.320 |
| *H. pylori* | 27 (75.0) | 11 (57.9) | 0.192 |
| Blood-type with B alleles | 11 (30.6) | 6 (31.6) | 0.938 |
| *MUC4* variants | 3 (8.3) | 14 (73.7) | **<0.001** |
| Histology of cancer | | | |
| Intestinal type | | 13 (63.2) | |
| Diffuse type[c] | | 4(26.3) | |
| Unknown | | 2 (10.5) | |
| HDGC[d] | | 3 (15.8) | |

Most values are shown as numbers (%) except for age which is expressed as a mean (standard deviation).

Bold font indicates statistical significance.

Abbreviations: GC, gastric cancer; HDGC, hereditary diffuse gastric cancer syndrome.

[a] Statistical significance was determined by the chi-squared test or t-test.

[b] Based on Lauren classification.

[c] Including 1 mixed type.

[d] Families with 2 or more cases of gastric cancer with at least 1 diffuse gastric cancer diagnosed before the age of 50 years old.

plots for LOD *p* values of three different gene size groups, and they confirm that our analyses are not affected by gene size.

## *MUC4* variants

When the full dataset was analyzed using pVAAST, 14 variants of *MUC4* were found to contribute to the LOD score, and 10 of these were finally selected with LOD values greater than 0. The 10 variants of *MUC4* were identified among 14 independent families (Table 2). All subjects who harbored these variants were affected by GC except for cases #16, #37 and #39 (Table 2). GC patients with *MUC4* variants mostly had intestinal-type GC except #23 (Fig 1). The three unaffected participants (#16, #37 and #39) were all female and nonsmokers. Two variants, namely, c.7658C>T p.A2553V and c.5005A>G p.S1669G, were identified in three unrelated families. Participants #15, #16, #19, #34 and #51 carried two different variants each. Most of the variants identification rates in populations of East Asian origin were higher than those observed in the overall population (Table 2).

## Effect size of *MUC4* variants in the development of gastric cancer

Standard logistic regression was applied to all 55 family members with adjustments for sex, tobacco smoking, and HDGC (see methods). Carrying any *MUC4* variation was associated with an increased risk of GC (*MUC4*; OR 58.08, 95% CI 7.33 to 459.97; online S3 Table).

## A. Results of pVAAST

| Transcript | Gene | LOD | *p* value of LOD | CLRT | *p* value of CLRT |
|---|---|---|---|---|---|
| >NM_018406.6 | MUC4 | 4.16 | $1.85 \times 10^{-05}$ | 106.6 | $<9.9 \times 10^{-09}$ |
| >NM_005462.4 | MAGEC1 | 2 | 0.000112 | 33.9 | $4.00 \times 10^{-06}$ |
| >NM_017750.3 | RETSAT | 1.6724 | 0.000126 | 38.985 | $9.00 \times 10^{-06}$ |

## B. LOD *p* values from pVAAST

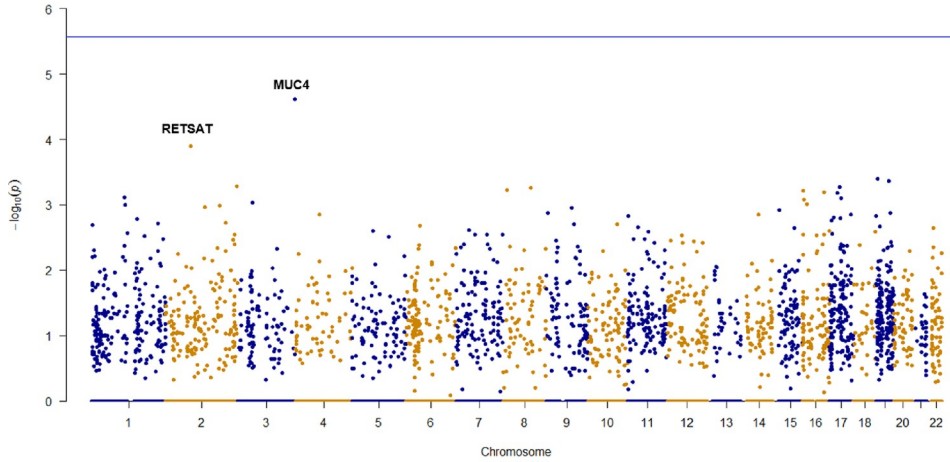

## C. QQ plot of LOD *p* values from pVAAST

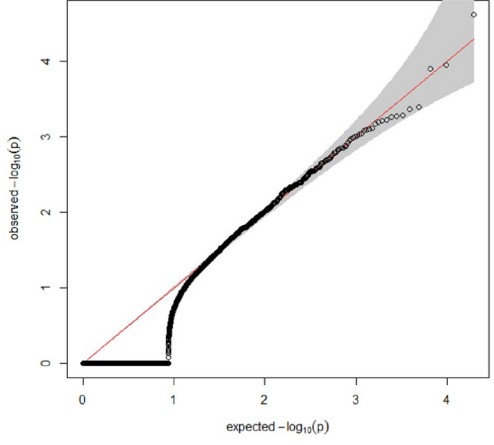

**Fig 2. Candidate predisposition genes for gastric cancer by combined linkage analyses with pVAAST.** (A) Predisposition genes that were detected are described. A composite likelihood ratio test (CLRT) based on the binomial likelihood for allele counts in cases and controls weighted by the functional prediction likelihood ratio was conducted with $10^6$ permuted samples by the gene drop method. (B) Manhattan plot of the logarithm of odds (LOD) *p* values of all protein-encoding genes from the pVAAST run; each dot in the plot represents a *p* value for one gene. The x-axis shows the genomic locations arranged by chromosome. (C) quantile-quantile (QQ) plot of the LOD *p* values from pVAAST.

**Table 2. Characteristics of the germline *MUC4* variants associated with gastric cancer by linkage analysis in the 14 studied families.**

| Location[a] | Allelic change | | AA change | LOD | Affected case | Population MAF | Exon | Functional prediction |
|---|---|---|---|---|---|---|---|---|
| | Ref | Alt | (NM_018406) | | (patient IDs)[b] | (overall/East Asian, %)[c] | | |
| chr3:195513076 rs774527434 | C | T | p.Arg1792His | 0.60 | **51**, **52**, **53** | 0.044/ 0.433 | 2 | O-glycosylation (Alpha subunit) |
| chr3:195513446 rs534579185 | T | C | p.Ser1669Gly | 1.20 | **34**, **2**, **30** | 0.262/ 1.460 | 2 | O-glycosylation (Alpha subunit) |
| chr3:195510793 rs77250903 | G | A | p.Ala2553Val | 1.57 | **11**, **19**, **23** | 0.062/ 0.028 | 2 | O-glycosylation (Alpha subunit) |
| chr3:195507271 rs868067409 | G | C | p.Thr3727Ser | 0.90 | **51** | 0.017/ 0.226 | 2 | O-glycosylation (Alpha subunit) |
| chr3:195475923 rs531395109 | G | A | p.Thr5295Met | 0.60 | **34**,37, **38**, 39 | 0.001/ 0.011 | 24 | N-glycosylation (Beta subunit) |
| chr3:195507778 rs754808151 | G | A | p.Ala3558Val | 0.35 | **5** | 0.016/ 0.000 | 2 | O-glycosylation (Alpha subunit) |
| chr3:195512387 rs1304612772 | G | A | p.Leu2022Phe | 0.30 | **32**,33 | 0.001/ 0.009 | 2 | O-glycosylation (Alpha subunit) |
| chr3:195510803 rs774907241 | G | T | p.Pro2550Thr | 0.30 | **19** | 0.008/ 0.000 | 2 | O-glycosylation (Alpha subunit) |
| chr3:195511813 rs771925912 | C | T | p.Ser2213Asn | 0.14 | **15**, **16** | 0.001/ 0.009 | 2 | O-glycosylation (Alpha subunit) |
| chr3:195511811 rs745342765 | G | T | p.Pro2214Thr | 0.04 | **15**, **16** | 0.000/ 0.000 | 2 | O-glycosylation (Alpha subunit) |

[a] Chromosome position in reference genome, GRCh37/hg19.

[b] ID of subjects diagnosed with gastric cancer is highlighted in a bold style and case numbers with an underline belong to the same family.

[c] MAF (%) in overall and East Asian population from gnomAD (v2.1) exome database.

## Validating the association of *MUC4* with gastric cancer in large case control cohort

As we hypothesized that hereditary and sporadic gastric cancer may share genetic background for gastric cancer, we further analyzed whether variants of *MUC4* are related to the development of GC in a large cohort which consists of 597 GC patients and 9759 healthy controls genotyped with SNP array. Common SNPs of *MUC4* regions (chr3: 195,473,637–195,539,149), including 0.5 MB of flanking region, were analyzed, and the Bonferroni-adjusted 0.05 significance level was $1.18 \times 10^{-5}$. Two common variants in *MUC4* lesions (rs148735556 and rs11717039) were detected (S3 Fig), suggesting the association of *MUC4* with GC. In exon 2 and 24 regions of *MUC4*, there were 25 SNPs, and the rs547775645 missense variant in exon 2 was identified to be significant at the $0.05/25 = 2 \times 10^{-3}$ significance level. The imputation quality of the above mentioned ten rare variants in *MUC4* was poor (INFO < 0.5), and they could not be tested in this SNP chip analysis. However, the presence of common variants in *MUC4* with significant association with GC supports that germline *MUC4* variants might be linked with GC.

## Frequency of *MUC4* germline variants in patients with multiple cancer types

We investigated the allele frequency of MUC4 variants in germline samples of patients with multiple cancer types because MUC4 is highly expressed not only in stomach but also other

tissues such as colon, esophagus, small intestine, uterus, and lung and patients with germline MUC4 variants might be at higher risk of developing multiple types of cancers. We tested whether the 10 rare variants of *MUC4* gene were related to cancer using germline variants from blood in the Cancer Genome Atlas Study data. We identified three rare variants in *MUC4* associated with gastric cancer and 4 other cancer types (online S4 Table). Out of 10 associated variants of the *MUC4* gene that have been linked to familial GC in Table 2, a hetero-zygote rs774527434 SNP was identified in one (0.17%) patient among 295 stomach adenocar-cinoma germline samples (online S4 Table), approximately 4 times higher than that in the general population (0.04%, Table 2). Two variants, rs534779185 and rs77250903, were identi-fied in 372 CRC patients, with frequencies of 4.0% and 0.13%, respectively, higher than that in the general population (0.26 and 0.06% respectively, Table 2). One variant, rs534779185, was found in 265 uterine corpus endometrial cancer samples with a frequency of 0.56%. It is inter-esting that none of the ten variants of the *MUC4* gene were identified in 408 lung squamous cell cancer or 495 lung adenocarcinoma patients, suggesting that *MUC4* variants might be related with gastrointestinal or genitourinary tract cancers.

## *MUC4* expression in stomach tissues of subjects with *MUC4* variants

To investigate the functional effects of the identified variants, *MUC4* expression in gastric mucosa was measured using IHC. Representative immunohistochemical results of 5 partici-pants in family No. 14 which showed complete cosegregation with *MUC4* variants (rs774527434) and contained the largest number of gastric cancer patients are shown (Fig 3). While *MUC4* variant-negative, noncancerous mucosa (#50, #54) (Fig 3A and 3D) displayed high intensities, the IHC results of *MUC4* variant-positive noncancerous mucosa from three patients (#51, #52 and #53; Fig 3B, 3E and 3H) with *MUC4* variants were weak or negative. In contrast, the cancer tissues of three patients (#51, #52 and #53; Fig 3C, 3F and 3I) showed high IHC scores.

Generally, noncancerous mucosa with *MUC4* variants exhibited a lower score of *MUC4*-positive staining than those with wild type (Fig 3G; median [interquartile range]: 0 [10.0–30.0] vs. 70 [9.5–165.0], $p = 0.023$). In cancer tissue, there was a tendency towards more prominent IHC staining in those with *MUC4* variants compared to the wild type (Fig 3G; median [inter-quartile range]: (75.0 (0–240.0) vs. 30 (0–105.0), $p = 0.287$)

## Prediction of protein structure

Nine of the 10 *MUC4* variants identified in the present study were located in exon 2, which includes a tandem repeat region [34], while the other variant existed in exon 24. Neither homology modeling nor ab initio structural modeling was successful due to the absence of established *MUC1* or *MUC4* models and the coil structure of the O-glycosylation-rich site.

According to the prediction for glycosylation [29, 35], most *MUC4* variants in exon 2 were O-glycosylation sites or physically close to them (online S5 Table). In particular, p.T3727S, p.S1669G and p.S2213N were more likely to carry O-GalNAc modifications [29]. Change in a single codon from G to A (c.5375G>A:p.R1792H) of this putative cleavage site may potentially inhibit proteolytic activity. The change in a single codon from C to T (c.15884C>T) hampered the synthesis of threonine, a prospective N-glycosylation site between the second and third epi-dermal growth factor (EGF)-like domains of the *MUC4* ß subunit (S4 Fig).

## Discussion

To identify novel GC-susceptible genes, we performed WES in 19 GC-affected members and 36 unaffected FDRs of 14 families in which 2 or more GC cases had occurred within the third

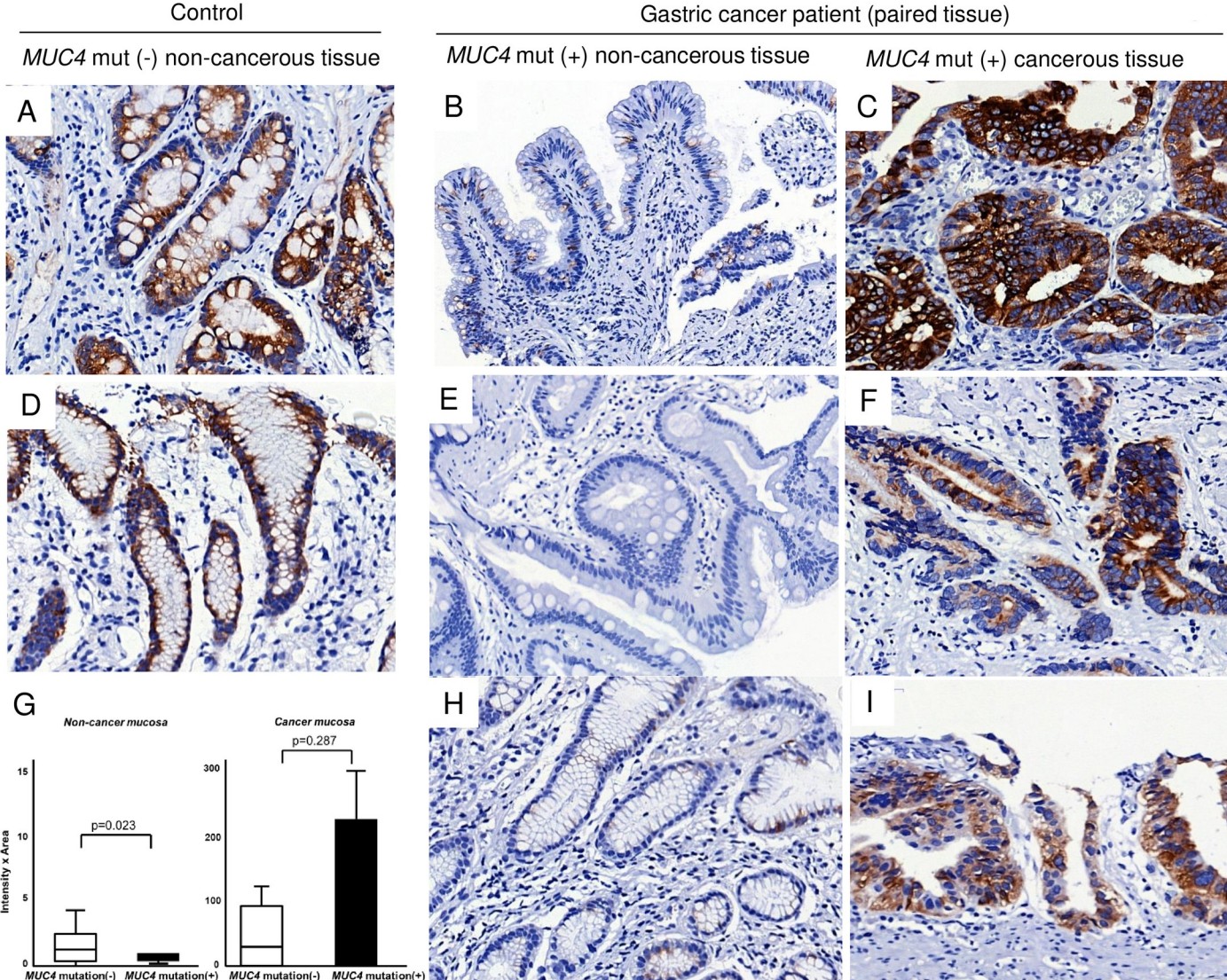

**Fig 3.** Representative photomicrographs of IHC for *MUC4* in noncancerous and cancerous gastric mucosal tissue of Family No. 14 (original magnification A-F, H, and I ×400). Panel A is tissue of #50 (participant number); B & C #51; D, #54; E & F, #52; H & I, #53. Intensive staining (brown) of *MUC4* in noncancerous tissue (A, D) in the representative *MUC4* variant-negative control is shown compared to absent or faint immunoreactivity of noncancerous tissue in *MUC4* variant-positive gastric cancer patients (B, E & H). The paired cancer tissue (C, F and I) of B, E and H show intense and diffuse staining of *MUC4*. G, *MUC4* immunointensity in *MUC4* variant-positive noncancerous mucosa was weaker than that in the *MUC4* variant-negative (left). Despite an increased tendency of immunointensity in *MUC4* variant-positive cancer tissue, no statistical significance was observed (right). The white bar represents *MUC4* variant-negative, while the black bars denote *MUC4* variant-positive.

generation. Linkage and association analyses identified *MUC4* missense variants as a predisposition to the familial aggregation of GC. The discovery of common variants in *MUC4* region significantly linked with gastric cancer in a large case control cohort of gastric cancer supports the association of *MUC4* with gastric cancer. Downregulated IHC results in normal gastric mucosa of *MUC4* variant carriers were found relative to wild type, suggesting the loss of protective function of *MUC4* in carriers of *MUC4* variants.

Unlike previous studies that recruited only hereditary diffuse gastric cancers, we invited families with more than two members with gastric cancer regardless of pathologic subtypes as our focus was to uncover GC susceptibility variants related to familial clustering. We invited

both affected and unaffected family members but it was very difficult to recruit relatives because some had already died from gastric cancer and others were reluctant to be taken blood or consent to genetic study. The practical difficulty in recruiting participants may be one factor that hampers the genetic study of cancers with familial clustering.

*MUC4* is a large, heavily glycosylated transmembrane mucin, varying from 550 to 930 kDa, due to the polymorphic variable number tandem repeat region [34]. It is expressed in the normal epithelium of several organs, including the stomach, intestine and mammary glands, and protects and lubricates the epithelium [34]. *MUC4* shares structural similarities with *MUC1* but also possesses unique regions, including three EGF-like domains, which are required for activation of ErbB2.

While the association *MUC1* and gastric carcinogenesis is relatively well documented [12], knowledge concerning the role of *MUC4* in gastric carcinogenesis is limited. Normal *MUC1* blocks adhesion of *H. pylori* to the gastric mucosa, thus preventing *H. pylori* colonization [36] and gastritis [37]. However, there was no significant interaction between H. pylori status and *MUC4* variants in the development of GC in our study. It might be due to the small sample size of the study cohort but a study with larger sample size that compared the clinical factors between GC patients with or without family history of GC suggested that the effect of H. pylori infection on GC development decreases among GC patients with family history of GC [7]. Two recent meta-analysis studies have shown that the G allele of *MUC1* rs4072037 is associated with a decreased risk of GC [38, 39]. *MUC1* rs4072037 affects the gene promoter, leading to reduced transcriptional activity and *MUC1* [40, 41].

In the present study, we attempted to explain the possible mechanism of the association between *MUC4* variant and GC risk through expression analyses of *MUC4*. Although only a limited number of gastric mucosal samples could be obtained, we found that *MUC4*-stained cells tended to decrease in the noncancerous gastric mucosa. As *MUC4* is structurally similar to *MUC1* and as carriers of *MUC4* variants displayed decreased expression of *MUC4*, we hypothesized that *MUC4* variants cause a detrimental effect by preventing the expression of *MUC4* in normal mucosa. This trend was most prominent among family members with c.5375G>A:p.R1792H of *MUC4* (Family No. 14) (Fig 3).

The structures of either MUC1 or MUC4 have not been experimentally confirmed to date. To characterize the molecular nature of *MUC4* where variants affect, we tried to predict protein structure. MUC4 contains 5412 amino acid residues. Among them, 4264 amino acid residues from the N-terminus are predicted to be an intrinsically disordered region (IDR) by the XtalPred server [42] that plays physiological roles through nonstructured domains [43]. Our attempts to model the exon 2 region of MUC4, where many variants are located, were unsuccessful because the N-terminal IDR of MUC4 does not contain templates for modeling.

Nevertheless, the information we obtained from *in silico* prediction of glycosylation in MUC4 structure indicates that most regions encoded by *MUC4* variants are likely to be O-glycosylation sites. O-glycosylation with glycan micro-heterogeneity is crucial to mucin structure and function. Mucin-type glycans are involved in specific ligand-receptor interactions and can confer hydroscopic properties and bind various small molecules and proteins, finally stabilizing the protein structure [44]. Although hundreds of O-glycosylation sites exist in the MUC4α subunit, a different amino acid on one specific site may lead to altered functioning of the encoded variant protein [45]. Positional preference for amino acids [46] and changes in serine or threonine [34] around the O-glycosylation site in the present study provide the basis of altered glycosylation of MUC4.

While mucins play an essential role in forming a protective barrier over various tissues under normal physiological conditions, abnormal *MUC4* expression has been reported in certain types of carcinomas, including those of the lung, breast, pancreas, and stomach [34]. The

IHC findings of the present study confirm those of previous studies that reported increased *MUC4* expression in cancer tissue compared to normal tissue (Fig 3). The excessive expression of *MUC4* in cancer tissue could reflect a dual role as an oncogene. Previous studies suggested that *MUC4* may activate the ErbB2 oncoprotein during the pathogenesis of GC [47, 48]. In addition, the variant in exon 24, p.Thr5295Met, might be involved in ErbB2 signaling because the variant causes an amino acid change in an N-glycosylation site between the EGF-like domains (S4 Fig). The structural model of a putative third EGF-like domain of MUC4 we managed to obtain covers residues from F5300 to L5362, which is in close proximity to variant residue T5295M. Because this site is only five residues away from the modeled EGF-like domain, it is plausible that this variant may affect the function of the EGF-like domain (S4 Fig).

The frequencies of the *MUC1* rs4072037 A allele in our study cohort of 90.9% were comparable to those in 1,124 Chinese GC patients (97.2%) [49], while those of the A and G alleles were 57.9% and 42.1% in American populations and 49.2% and 50.8% in African populations, respectively [50]. These results indicate that East Asian populations may be genetically susceptible to GC through *MUC1* variants. The frequencies of most identified *MUC4* variants in the present study were generally higher in the East Asian population than those observed in the global population, including those of western origin (Table 2). Moreover, One variant of *MUC4* (rs774527434) was identified in Asian GC patient in TCGA. Overall, putative *MUC4* variants in this study may contribute to geographic differences in GC incidence parallel to *MUC1* variant.

As GC has a heterogeneous etiology, individuals in the GC family could exhibit a discrepancy between genetic susceptibility and clinical presentation. In the present study, 5 GC patients without *MUC4* variants were identified among 4 independent families: #20, #28, #43, #45, and #46 (Fig 1). Diffuse-type GC patient #28, who met the HDGC criteria, harbored novel missense mutations in *CDH1* (NM_001317184: exon8:c.G1057A:p.E353K). Although family #46 met the HDGC criteria, no mutation in *CDH1* or *CTNNA1* was found [17, 51]. Patients #43 and #45 belonged to the same family with two renal cell cancers, suggesting the possibility of other genetic syndromes. Notably, the association of *MUC4* variants with GC development may be strong in non-HDGC and mostly intestinal-type GC in the Korean population.

The present study did not conduct functional analyses for *MUC4*. Nonetheless, the strength of the present study is that a family-based linkage analysis was designed in a familial clustering setting. The presence of affected and unaffected members in a family allowed for the selection of variants whose phenotypes segregated on a per-family basis. We provided external validation data with SNP chip analysis in large case control cohort of GC. We also included intestinal-type GC, while many studies on familial GC have shown that genetic predisposition correlated with diffuse-type GC only.

The limitations of our study are that the sample size was small and we could not experimentally demonstrate the functional significance of MUC4 variants. And we could not adjust for confounders or examine the difference in demographic characteristics as the whole genome dataset of 379 Koreans from the National Biobank of Korea do not contain phenotype data and we did not investigate genomic structural variations such as large copy number variations. Also, we could not investigate the association of MUC4 variants with disease severity as some of the patients with gastric cancer were treated at other hospitals and clinical data about disease severity like tumor staging and survival were not available.

## Supporting information

**S1 Fig. Motif search was performed using MotifFinder tool.**
(PDF)

**S2 Fig. QQ plots for LOD p values of three different gene size groups.**
(PDF)

**S3 Fig. GWAS analysis.**
(PDF)

**S4 Fig. A predicted structure of T5295M (chr3:195475923) using structure prediction tool Phyre2.** T5295M encoding region was a loop region between the 2nd and 3rd EGF domain of MUC4 $\alpha$. Especially, the third EGF like domain of MUC4 was modeled by Modeller 9v10. Chain D from LRP4 complex structure (PDB Id: 3v65) was used as a template for homology modelling which was searched by Swissmodel server. The SNP loci is also near or part of N glycosylation cite with the N glycosylation pattern of N Xaa ST→ML. The red sphere denotes SNP loci and predicted EGF domains are color coded in blue, sky blue, and orange.
(PDF)

**S1 Table. Characteristics of 55 participants in the present study.**
(PDF)

**S2 Table. Summary of whole exome sequencing quality control statistics.**
(PDF)

**S3 Table. Odds ratios of independent risk factors for gastric cancer by binary logistic regression among participants of the present study.**
(PDF)

**S4 Table. Frequencies of 3 SNPs in multiple tumor types from The Cancer Genome Atlas Study.**
(PDF)

**S5 Table. Glycosylation site prediction using NetOGlyc and NetNGlyc.**
(PDF)

## Author Contributions

**Conceptualization:** Nayoung Kim, Sungho Won.

**Data curation:** Yoon Jin Choi, Jung Hun Ohn, Nayoung Kim, Seung Joo Kang, Joo Sung Kim.

**Formal analysis:** Yoon Jin Choi, Jung Hun Ohn, Wonji Kim, Kyungtaek Park, Sungho Won, Lee Sael, Cheol Min Shin, Sun Min Lee, Sejoon Lee, Hyun Joo An, Dong Man Jang, Byung Woo Han, Hye Seung Lee, Seung Joo Kang.

**Funding acquisition:** Nayoung Kim.

**Investigation:** Yoon Jin Choi, Jung Hun Ohn, Nayoung Kim, Wonji Kim, Kyungtaek Park, Sungho Won, Lee Sael, Cheol Min Shin, Sun Min Lee, Sejoon Lee, Hyun Joo An, Dong Man Jang, Byung Woo Han, Hye Seung Lee, Joo Sung Kim, Dong Ho Lee.

**Methodology:** Yoon Jin Choi, Jung Hun Ohn, Nayoung Kim, Kyungtaek Park, Sungho Won, Cheol Min Shin, Sun Min Lee, Sejoon Lee, Hyun Joo An, Dong Man Jang, Byung Woo Han, Hye Seung Lee, Seung Joo Kang, Joo Sung Kim, Dong Ho Lee.

**Project administration:** Nayoung Kim.

**Resources:** Nayoung Kim.

**Software:** Sejoon Lee.

**Supervision:** Nayoung Kim.

**Writing – original draft:** Yoon Jin Choi, Jung Hun Ohn, Nayoung Kim, Kyungtaek Park, Sungho Won, Lee Sael, Cheol Min Shin, Sejoon Lee.

**Writing – review & editing:** Yoon Jin Choi, Jung Hun Ohn, Nayoung Kim.

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
