## [Decision Letter · Decision Letter 0]

9 Jun 2020

PONE-D-20-15912

Family-based exome sequencing combined with linkage analyses identifies rare susceptibility variants of MUC4 for gastric cancer

PLOS ONE

Dear Dr. Nayoung Kim,

Thank you for submitting your manuscript to PLOS ONE. this manuscript was evaluated by two reviewer who has extensive knowledge and expertise in the related field. Both reviewers agree that this study is well designed and executed. The reviewers have several minor concerns and questions for this manuscript. We invite you to submit a revised version of the manuscript that addresses the points raised by the reviewers.  

If the authors choose to revise the paper, the revised version will need to go back to the reviewers for reassessment. Specific comments from the reviewer are provided below and each should be carefully taken into consideration.

We encourage you to submit your revision within forty-five days of the date of this decision.  Please include the following items when submitting your revised manuscript:

We look forward to receiving your revised manuscript.

Kind regards,

Seungil Ro, PhD

Academic Editor

PLOS ONE

Journal Requirements:

Additional Editor Comments (if provided):

Reviewers' comments:

Reviewer's Responses to Questions

**Comments to the Author**

1. Is the manuscript technically sound, and do the data support the conclusions?

Reviewer #1: Yes

Reviewer #2: Yes

2. Has the statistical analysis been performed appropriately and rigorously? 

Reviewer #1: Yes

Reviewer #2: Yes

3. Have the authors made all data underlying the findings in their manuscript fully available?

Reviewer #1: Yes

Reviewer #2: Yes

4. Is the manuscript presented in an intelligible fashion and written in standard English?

Reviewer #1: Yes

Reviewer #2: Yes

5. Review Comments to the Author

Reviewer #1: Reviewer’s comment on “Family-based exome sequencing combined with linkage analyses identifies rare susceptibility variants of MUC4 for gastric cancer” (PONE-D-20-15912) authored by Choi et al.

This study aimed to identify germline variants predisposing for gastric cancer by using linkage analysis and association analysis. Authors conducted whole exome sequencing based on DNA from blood samples of 19 gastric cancer patients and 36 unaffected family members from 14 families. The result of their linkage analysis identified MUC4 as a predisposing gene for gastric cancer. A further genetic association analysis based on 597 gastric cancer patients and 9,759 normal controls suggested three SNPs on MUC4 associated with gastric cancer. Immunohistochemistry experiment suggested the loss of protective function of MUC4 for the carriers of germline missense mutations in MUC4. In general, the paper is well written and the results are informative.

Abstract. “And the MUC4 variants were found in higher frequency in The Cancer Genome Atlas Study (TCGA) germline samples of patients with multiple cancer types.” Is it not clear why authors would like to examine the rare variants of MUC4 in other cancers in TCGA? In addition, when the allele frequency was compared, did authors consider the difference in allele frequency for the individuals with different ethnic background? In the comparison, the variant data were all from blood samples or from different tissue types? In addition, on page 19, the description “One variant of MUG …” seems not informative and cannot support the statement in abstract. The materials related to TCGA may be removed from this study.

Materials and method. Authors compared allele counts in whole exome of 19 gastric cancer patients to those of 397 Korean control from National Biobank of Korea. The clinical and demographic characteristics were similar? Potential confounders were adjusted for in the comparison?

Authors are suggested to examine structural variation at least for MUC4 in gastric cancer patients and healthy controls.

Reviewer #2: Minor comments:

In the present study (PONE-D-20-15912) the authors explored to identify novel gastric cancer (GC)-susceptible genes and performed whole-exome sequencing (WES) in 19 GC-affected members and 36 unaffected FDRs of 14 families in which 2 or more GC cases had occurred within the third generation. Linkage and association analyses identified MUC4 missense variants as a predisposition to the familial aggregation of GC. Although, this is a large and well-designed case-control study and authors have performed extensive tools to discover common variants in MUC4 region which are significantly linked with GC, I have following minor comments regarding the study:

1. Sample size calculation for patients with GC and control is not mentioned. For calculation of sample size, which software was used and specify the test used. How the power of the study was calculated? Also, please mention how the effect size of SNPs was analyzed?

2. Inclusion and exclusion criteria for the participants should be mentioned clearly. For example, how the healthy subjects were recruited? Did they fill out questionnaires on gastrointestinal symptoms? For a diagnosis of GC, was it based on one or a combination of clinical, radiological, and endoscopic criteria?

3. Is the genotype frequency in MUC4 gene differ in any particular age group?

4. H. pylori status in co-relation with MUC4 variant might have some roles in pathogenesis of GC. The results section lacks association between GC with or without MUC4 variant and H. pylori status. Also, some discussion on this point needs to be included.

5. In the introduction section, it should be mentioned on the role of MUC on host physiology, and how mutant variant might affect in the pathogenesis. The roles of MUC4 can only be speculative at this moment without any functional studies.

6. Please mention the clinical history of patients with anti-microbial therapy, H2-receptor blockers etc. were excluded from this study? This will make the study for better presentation.

7. How the variant MUC4 is associated with GC disease severity?

8. Did the authors observe any association of MUC1, and MUC4 variants in a same GC patient? As MUC1variant and gastric carcinogenesis is well documented.

9. What are the study limitations? Addition of a paragraph on this would be better.

10. Though the host's genetic factors play an important role in gastric carcinogenesis, the role of diet cannot be denied. History of dietary habit might add some novel finding in relation with MUC4 variant.

6. PLOS authors have the option to publish the peer review history of their article (what does this mean?). If published, this will include your full peer review and any attached files.

Reviewer #1: No

Reviewer #2: No

---

## [Author Response · Author response to Decision Letter 0]

18 Jun 2020

Dear Editor-in-Chief: 

RE: PONE-D-20-15912

Family-based exome sequencing combined with linkage analyses identifies rare susceptibility variants of MUC4 for gastric cancer

coauthored by Yoon Jin Choi, Jung Hun Ohn, Sung Ho Won, Won Ji Kim, Kyungtaek Park, Lee Sael, Cheol Min Shin, Sun Min Lee, Sejoon Lee, Hyun Joo An, Dong Man Jang, Byung Woo Han, Hye Seung Lee, Seung Joo Kang, Joo Sung Kim and Dong Ho Lee.

Thank you very much for giving us the opportunity for revision. Accurate and kind comments by the reviewer have been addressed in the revised manuscript. We also believe that these comments improved our manuscript. Major changes have been highlighted in Red color in the revised manuscript, and revised supplementary table 1 to avoid any confusion. 

I anticipate good response.

Thank you!

Sincerely,

Nayoung Kim, M.D.

Reviewer #1: 

Reviewer’s comment on “Family-based exome sequencing combined with linkage analyses identifies rare susceptibility variants of MUC4 for gastric cancer” (PONE-D-20-15912) authored by Choi et al.

This study aimed to identify germline variants predisposing for gastric cancer by using linkage analysis and association analysis. Authors conducted whole exome sequencing based on DNA from blood samples of 19 gastric cancer patients and 36 unaffected family members from 14 families. The result of their linkage analysis identified MUC4 as a predisposing gene for gastric cancer. A further genetic association analysis based on 597 gastric cancer patients and 9,759 normal controls suggested three SNPs on MUC4 associated with gastric cancer. Immunohistochemistry experiment suggested the loss of protective function of MUC4 for the carriers of germline missense mutations in MUC4. In general, the paper is well written and the results are informative.

Abstract. “And the MUC4variants were found in higher frequency in The Cancer Genome Atlas Study (TCGA) germline samples of patients with multiple cancer types.” Is it not clear why authors would like to examine the rare variants of MUC4 in other cancers in TCGA? 

Response>

Thank you very much for the comment. We investigated the allele frequency of MUC4 variants in germline samples of multiple cancer types because MUC4 is highly expressed not only in stomach but also other tissues such as colon, esophagus, small intestine, uterus, and lung (https://www.proteinatlas.org/ENSG00000145113-MUC4/tissue) and patients with germline MUC4 variants might be at higher risk of developing cancers in those tissues. Following the reviewer’s comment, we added the following sentence to explain why we examined the allele frequency of MUC4 variants in multiple cancer types.

Results section page 14 lines 20-23>

“We investigated the allele frequency of MUC4 variants in germline samples of patients with multiple cancer types because MUC4 is highly expressed not only in stomach but also other tissues such as colon, esophagus, small intestine, uterus, and lung and patients with germline MUC4 variants might be at higher risk of developing multiple types of cancers.”

In addition, when the allele frequency was compared, did authors consider the difference in allele frequency for the individuals with different ethnic background? 

Response>

We really appreciate the reviewer’s comment. Unfortunately, we could not take into the difference in allele frequency with different ethnic background as only 3% of TCGA patients are Asians as reported by Spratt DE et al. (JAMA Oncol. 2016 Aug 1;2(8):1070-4.) Therefore, we had to compare the allele frequency to that in the general population.

In the comparison, the variant data were all from blood samples or from different tissue types? 

Response>

The variant data were all from blood samples. We specified that the variant data were from blood samples in the results section as following:

Page 14, line 24>

“We tested whether the 10 rare variants of MUC4 gene were related to cancer using germline variants from blood in the Cancer Genome Atlas Study data”

In addition, on page 19, the description “One variant of MUC4 …” seems not informative and cannot support the statement in abstract. The materials related to TCGA may be removed from this study.

Response>

We appreciate the specific suggestion. As suggested by the reviewer, we removed the TCGA related sentence from the abstract. But we did not totally remove the TCGA part as we thought that the analysis may be informative because MUC4 is highly expressed not only in stomach but also other tissues such as colon, esophagus, small intestine, uterus, and lung (https://www.proteinatlas.org/ENSG00000145113-MUC4/tissue) and patients with germline MUC4 variants might be at higher risk of developing cancers in those tissues.

Materials and method. Authors compared allele counts in whole exome of 19 gastric cancer patients to those of 397 Korean control from National Biobank of Korea. The clinical and demographic characteristics were similar? Potential confounders were adjusted for in the comparison?

Response>

We are thankful for the comment. As the whole genome dataset of 379 Koreans from the National Biobank of Korea do not contain phenotype data, we could not adjust for confounders or examine the difference in demographic characteristics. We added the point in the limitation section of the discussion.

Page 20, lines 1-3 > 

“The limitations of our study are that … And we could not adjust for confounders or examine the difference in demographic characteristics as the whole genome dataset of 379 Koreans from the National Biobank of Korea do not contain phenotype data …”

Authors are suggested to examine structural variation at least for MUC4 in gastric cancer patients and healthy controls.

Response>

We really appreciate the suggestion. Calling structural variation such as copy number deletions from exome sequencing dataset is limited although tools like EXCAVATOR are available. We plan to investigate exome-wide structural variation from our whole exome dataset of gastric cancer families in a separate study and added it in limitation section of the discussion.

Page 20 lines 3-4>

“The limitations of our study are that … and we did not examine structural variations such as large copy number variations.”

The authors really appreciate the reviewer’s kind and accurate comments. The revision based on these comments made this manuscript more accurate and the quality improved. Thank you. 

Nayoung Kim, M.D., Ph.D.

 

Reviewer #2: Minor comments:

In the present study (PONE-D-20-15912) the authors explored to identify novel gastric cancer (GC)-susceptible genes and performed whole-exome sequencing (WES) in 19 GC-affected members and 36 unaffected FDRs of 14 families in which 2 or more GC cases had occurred within the third generation. Linkage and association analyses identified MUC4 missense variants as a predisposition to the familial aggregation of GC. Although, this is a large and well-designed case-control study and authors have performed extensive tools to discover common variants in MUC4 region which are significantly linked with GC, I have following minor comments regarding the study:

1. Sample size calculation for patients with GC and control is not mentioned. For calculation of sample size, which software was used and specify the test used. How the power of the study was calculated? Also, please mention how the effect size of SNPs was analyzed?

Response>

We appreciate the comment. We did not calculate sample size before the study design. As we mentioned in the second paragraph of the discussion section, it is very difficult to recruit relatives because some had already died from gastric cancer and others were reluctant to be taken blood or consent to genetic study. The practical difficulty in recruiting participants may be one factor that hampers the genetic study of cancers with familial clustering. At first we recruited 10 families with familial clustering of gastric cancer, and we found significant signal in MUC4 variants. We tried to validate the result by recruiting 4 additional families and the result was consistent. 

We described how the effect size of MUC4 SNPs was calculated in page 9 of methods section as following:

Page 9, lines 16-23> 

“The odds ratio (OR) for all significant genes at the Bonferroni-adjusted 0.05 significance level was estimated with logistic regression. Familial correlations were estimated with GMMAT, and the variance for the random effect that explains familial correlation was estimated to be 0. Thus, GC status among family members was assumed to be independent, and standard logistic regression was applied using Rex Version 2.1 (http://rexsoft.org). For each gene, genetic risk scores were coded as 1 if one or more rare alleles in the corresponding gene were observed and 0 otherwise. Sex, age, smoking status and HDGC were included as covariates to adjust for their effects.”

2. Inclusion and exclusion criteria for the participants should be mentioned clearly. For example, how the healthy subjects were recruited? Did they fill out questionnaires on gastrointestinal symptoms? For a diagnosis of GC, was it based on one or a combination of clinical, radiological, and endoscopic criteria?

Response>

We are very thankful for the specific comment. Subjects in families with two or more members diagnosed with GC within three generations were asked to fill out questionnaires about family history of GC, smoking, consumption of alcohol, dietary preference, socioeconomic status, a history of previous H. pylori eradication and gastrointestinal symptoms. Healthy subjects were defined as individuals aged > 50 years with a normal endoscopy within the previous 6 months. For diagnosis of GC, it was based on pathologic diagnosis by endoscopic biopsy or surgical specimens. We added it in methods section page 6 line 6-9,

“From April 2017 to March 2018, GC patients and their FDRs, among families with two or more members diagnosed with GC within three generations, were enrolled in the study at Seoul National University Bundang Hospital. Non-GC controls were defined as individuals aged > 50 years with a normal endoscopy within the previous 6 months. For diagnosis of GC, it was based on pathologic diagnosis by endoscopic biopsy or surgical specimens.

Family history of GC, smoking, consumption of alcohol, dietary preference, socioeconomic status, gastrointestinal symptoms and a history of previous H. pylori eradication were acquired via questionnaires. Histologic evaluations with Giemsa staining and an anti-H. pylori test were performed to determine H. pylori infection status.”

3. Is the genotype frequency in MUC4 gene different in any particular age group?

Response>

We thank the reviewer for the specific comment. There was no difference in genotype frequency in MUC4 in age groups. And the mean age of subjects with MUC4 variants versus subjects without MUC4 variants were not significantly different (63.7 years vs. 60.0 years, p=0.209).

4. H. pylori status in co-relation with MUC4 variant might have some roles in pathogenesis of GC. The results section lacks association between GC with or without MUC4 variant and H. pylori status. Also, some discussion on this point needs to be included.

Response>

We appreciate the comment. There was no significant interaction between H. pylori status and MUC4 variants in the development of GC. It might be associated with the small sample size but in a study with large sample size that compared the clinical factors between GC patients with or without family history of GC suggest that the effect of H. pylori infection on GC development decreases among GC patients with family history of GC. (Medicine (Baltimore) 2016 May;95(20):e3606). We added the point in the discussion section page 17, lines 22-26.

“However, there was no significant interaction between H. pylori status and MUC4 variants in the development of GC in our study. It might be due to the small sample size of the study cohort but a study with larger sample size that compared the clinical factors between GC patients with or without family history of GC suggested that the effect of H. pylori infection on GC development decreases among GC patients with family history of GC[7].”

5. In the introduction section, it should be mentioned on the role of MUC on host physiology, and how mutant variant might affect in the pathogenesis. The roles of MUC4 can only be speculative at this moment without any functional studies.

Response>

We really appreciate the comment. With respect to MUC1 which belongs to the mucin family, it is located at the apical surface of the mucosal epithelial cells and acts as a protective barrier against exogenous insults. It is hypothesized that MUC1 variants like rs4072037 influence the quantity and the quality of the MUC1 protein and cause difference in barrier function in the stomach with subsequent difference in GC susceptibility between individuals (Int J Mol Sci. 2014 May 7;15(5):7958-73). We added the point in the introduction section of the revised manuscript page 4 lines 17-22, 

“One of the most well-known is the association of MUC1 with gastric cancer [12]. MUC1 belongs to the mucin family and it is located at the apical surface of the mucosal epithelial cells and acts as a protective barrier against exogenous insults. It is hypothesized that MUC1 variants like rs4072037 influence the quantity and the quality of the MUC1 protein and cause difference in barrier function in the stomach with subsequent difference in GC susceptibility between individuals [12].”

6. Please mention the clinical history of patients with anti-microbial therapy, H2-receptor blockers etc. were excluded from this study? This will make the study for better presentation.

Response> 

We appreciate the comment. Even though H. pylori was treated, authors thought that past infections could still affect the gastric carcinogenesis in some degree. In addition, there are many cases that do not have exact information about the anti-H. pylori therapy and the entire study subjects are small, authors could not do stratification analysis. We will conduct next research, taking into account the reviewer's comments. 

A summary of whether H. pylori eradicated was described in supplementary table S1 with the new Erad column. Thank you!

7. How the variant MUC4 is associated with GC disease severity?

Response>

We are thankful for the comment of the reviewer. Unfortunately, we could not investigate the association of MUC4 variants with disease severity as some of the patients with gastric cancer were treated at other hospitals and clinical data about disease severity like tumor staging and survival were not available. We added the point in the limitation paragraph of the revised manuscript page 20 lines lines 4-6. 

“ … Also, we could not investigate the association of MUC4 variants with disease severity as some of the patients with gastric cancer were treated at other hospitals and clinical data about disease severity like tumor staging and survival were not available.”

8. Did the authors observe any association of MUC1, and MUC4 variants in a same GC patient? As MUC1variant and gastric carcinogenesis is well documented.

Response>

We did not observe the association between MUC1 and MUC4 variants in the same GC patient. It may be because the well-documented “A” allele of the MUC1 variant rs4072037, that increases the risk of gastric cancer, was very common in our study cohort (the allele frequency was as high as 90.9% in our study cohort as described in the Discussion section page 19 line 16). 

9. What are the study limitations? Addition of a paragraph on this would be better.

Response>

We appreciate the helpful comment of the reviewer. We added a paragraph on the study limitations in the revised manuscript, page 19 line 28 to page 20 line 6, 

“The limitations of our study are that the sample size was small and we could not experimentally demonstrate the functional significance of MUC4 variants. And we could not adjust for confounders or examine the difference in demographic characteristics as the whole genome dataset of 379 Koreans from the National Biobank of Korea do not contain phenotype data and we did not investigate genomic structural variations such as large copy number variations. Also, we could not investigate the association of MUC4 variants with disease severity as some of the patients with gastric cancer were treated at other hospitals and clinical data about disease severity like tumor staging and survival were not available.” 

10. Though the host's genetic factors play an important role in gastric carcinogenesis, the role of diet cannot be denied. History of dietary habit might add some novel finding in relation with MUC4 variant.

Response>

Thank you very much for the suggestion. In our previous study, which included a cohort with large sample size, the intra-familial aggregation of gastric cancer was associated with environmental factors such as diet shared during childhood (Eur J Gastroenterol Hepatol. 2011;23:411-7) However, there was no significant association between dietary factors and gastric cancer in our study subjects, which might be because of the small sample size of the current study. 

The authors really appreciate the reviewer’s kind and accurate comments. The revision based on these comments made this manuscript more accurate and the quality improved. Thank you. 

Nayoung Kim, M.D., Ph.D.

---

## [Decision Letter · Decision Letter 1]

1 Jul 2020

Family-based exome sequencing combined with linkage analyses identifies rare susceptibility variants of MUC4 for gastric cancer

PONE-D-20-15912R1

Dear Dr. Nayoung Kim,

We’re pleased to inform you that your manuscript has been judged scientifically suitable for publication and will be formally accepted for publication once it meets all outstanding technical requirements.

Kind regards,

Seungil Ro, PhD

Academic Editor

PLOS ONE

Additional Editor Comments (optional):

Reviewers' comments:

Reviewer's Responses to Questions

**Comments to the Author**

1. If the authors have adequately addressed your comments raised in a previous round of review and you feel that this manuscript is now acceptable for publication, you may indicate that here to bypass the “Comments to the Author” section, enter your conflict of interest statement in the “Confidential to Editor” section, and submit your "Accept" recommendation.

Reviewer #1: All comments have been addressed

Reviewer #2: All comments have been addressed

2. Is the manuscript technically sound, and do the data support the conclusions?

Reviewer #1: Yes

Reviewer #2: Yes

3. Has the statistical analysis been performed appropriately and rigorously? 

Reviewer #1: Yes

Reviewer #2: Yes

4. Have the authors made all data underlying the findings in their manuscript fully available?

Reviewer #1: No

Reviewer #2: Yes

5. Is the manuscript presented in an intelligible fashion and written in standard English?

Reviewer #1: Yes

Reviewer #2: Yes

6. Review Comments to the Author

Reviewer #1: (No Response)

Reviewer #2: The manuscript (PONE-D-20-15912) entitled " Family-based exome sequencing combined with linkage analyses identifies rare susceptibility variants of MUC4 for gastric cancer" has been revised so well. Authors have clarified all the previous comments raised by the reviewer. They have extensively revised almost all sections of the manuscript. Also, authors have clarified reviewer’s comments. I do not have any other specific comment at this stage. Manuscript is in good shape and data are presented well. It may be acceptable.

7. PLOS authors have the option to publish the peer review history of their article (what does this mean?). If published, this will include your full peer review and any attached files.

Reviewer #1: No

Reviewer #2: No

---

## [Editor Report · Acceptance letter]

7 Jul 2020

PONE-D-20-15912R1 

Family-based exome sequencing combined with linkage analyses identifies rare susceptibility variants of MUC4 for gastric cancer 

Dear Dr. Kim:

I'm pleased to inform you that your manuscript has been deemed suitable for publication in PLOS ONE. Congratulations! Your manuscript is now with our production department. 

Kind regards, 

on behalf of

Dr. Seungil Ro 

Academic Editor

PLOS ONE